# Sex/Gender- and Age-Related Differences in β-Adrenergic Receptor Signaling in Cardiovascular Diseases

**DOI:** 10.3390/jcm11154280

**Published:** 2022-07-22

**Authors:** Daniela Liccardo, Beatrice Arosio, Graziamaria Corbi, Alessandro Cannavo

**Affiliations:** 1Department of Neurosciences, Reproductive and Odontostomatological Sciences, Federico II University of Naples, 80131 Naples, Italy; liccardo.daniela@gmail.com; 2Department of Clinical Sciences and Community Health, University of Milan, 20122 Milan, Italy; beatrice.arosio@unimi.it; 3Department of Medicine and Health Sciences, University of Molise, 86100 Campobasso, Italy; graziamaria.corbi@unimol.it; 4Department of Translational Medical Sciences, Federico II University of Naples, 80131 Naples, Italy

**Keywords:** cardiovascular disease, sex differences, β-adrenergic receptor, G protein-coupled receptors

## Abstract

Sex differences in cardiovascular disease (CVD) are often recognized from experimental and clinical studies examining the prevalence, manifestations, and response to therapies. Compared to age-matched men, women tend to have reduced CV risk and a better prognosis in the premenopausal period. However, with menopause, this risk increases exponentially, surpassing that of men. Although several mechanisms have been provided, including sex hormones, an emerging role in these sex differences has been suggested for β-adrenergic receptor (β-AR) signaling. Importantly, β-ARs are the most important G protein-coupled receptors (GPCRs), expressed in almost all the cell types of the CV system, and involved in physiological and pathophysiological processes. Consistent with their role, for decades, βARs have been considered the first targets for rational drug design to fight CVDs. Of note, β-ARs are seemingly associated with different CV outcomes in females compared with males. In addition, even if there is a critical inverse correlation between β-AR responsiveness and aging, it has been reported that gender is crucially involved in this age-related effect. This review will discuss how β-ARs impact the CV risk and response to anti-CVD therapies, also concerning sex and age. Further, we will explore how estrogens impact β-AR signaling in women.

## 1. Introduction

Cardiovascular diseases (CVDs) are the leading cause of mortality worldwide in men and women [1], although sex differences exist in the disease manifestation, prevalence, and outcome [2]. The scientific community generally agrees that women are more protected than men, although higher mortality rates and poorer prognoses are often observed in women with CVDs [2,3,4,5,6].

Several causal factors are behind these discrepancies, with the aging process and the activity of sex hormones such as estrogens (Es) as the primary culprits identified so far [7,8,9,10]. Indeed, in the premenopausal period, the incidence of CVDs is lower in females than in age-matched males [11,12,13,14]. However, as demonstrated by the Framingham study, with menopause, when the levels of Es decline, there is about a two-fold increase in CV event incidence compared to premenopausal women [13,14,15]. Furthermore, it is essential to underline that women are usually underrepresented in clinical studies [5,16]. Therefore, most of the data collected from preclinical and clinical studies lack crucial information about the differences between males and females. For this reason, better knowledge of sex-related differences, including the biological and physiological basis of these, is required to tailor CV therapeutic approaches.

This review explored the role of β-adrenergic receptors (β-ARs) and their related signaling pathways in males and females. βARs are the most important G protein-coupled receptor (GPCR) class expressed in whole CV cells, and they represent the most potent means to regulate cardiac and vascular function that respond to endogenous catecholamine hormones (epinephrine and norepinephrine) originating from the sympathetic nervous system (SNS) [17,18,19,20,21,22,23,24]. These receptors are implicated in controlling physiological processes such as blood pressure regulation, cardiac contractility, and metabolism within the CV system [17,18,19,20,21,22,23,24]. A dysregulation in β-AR responsiveness due to increased SNS activity, with consequent catecholamine bombardment of the receptors, is associated with several pathological conditions such as stroke, hypertension, cardiac hypertrophy, and heart failure (HF) [17,18,19,20,21,22,23,24]. For these reasons, β-ARs are considered the most important molecular targets in the CV system (i.e., β-blockers) [20].

Over the years, more and more studies have demonstrated a difference in β-AR responsiveness and signaling activation between males and females that may help to explain the sex differences in CVD. With this premise in mind, in this review article, we will discuss what is known about the role of β-ARs and how these impact CV physiology and pathophysiology. Then, we will summarize the experimental and clinical findings, showing how sex/gender differences and aging influence β-AR signaling and the response to different treatment strategies to fight CVDs and the patients’ outcomes.

## 2. β-Adrenergic Receptors and Cardiovascular System

The β-AR family consists of three subtypes (β1-, β2-, and β3-AR), whose expression (both mRNA and protein levels) and functions vary according to the cell types of the CV system [20,22,25,26,27,28,29,30,31]. For instance, in cardiomyocytes, all these receptors are expressed [25], but β1-AR is the highest expressed [26,27]. Conversely, in vascular smooth muscle cells (VSMCs) and cardiac fibroblasts (CFs), the β2-AR represents the prevalent subtype [25,28,29]; while in endothelial cells (ECs), β2- and β3-AR are the predominant receptors expressed [22,25]. In addition, previous studies have suggested the existence of a putative additional fourth subtype (β4-AR). The presence of β4-AR has been supported almost exclusively by results obtained from the aryloxypropanolamine agonist of β3-AR called CGP-12177 [32]. This molecule, developed as a β1/β2-AR antagonist, was shown to be a partial agonist of β3-AR (both native and recombinant), and studies in β3-AR knockout (KO) mice indicated that this fourth novel subtype was the primary mediator of the cardiac and thermogenic responses induced by CGP-12177 [32,33,34,35,36]. Therefore, these indirect experimental proofs suggested that β4-AR expression was presumably limited to tissues, such as fat and the heart. [32,37,38,39,40].

β-ARs belong to the superfamily of GPCRs, implying that their signaling is dependent on the coupling to the heterotrimeric G proteins (Gα-, Gβ-, and Gγ-subunits) and their binding to guanine nucleotides [20,22,30,31,41,42]. Notably, upon ligand binding, the activated β-AR serves as a guanine nucleotide exchange factor (GEF) for G proteins, catalyzing the release of bound guanosine diphosphate (GDP) and its replacement with guanosine triphosphate (GTP) on the α subunit [42]. This process leads to the dissociation of G proteins from the receptor and their separation into the Gα subunit and Gβγ dimers, subsequently activating several intracellular signaling cascades [22,42]. The specific function and signaling activated by each β-AR in the CV system are primarily defined by the Gα subunit to whom there are coupled. In this regard, stimulation of all β-ARs (β1-, β2-, β3-, and the putative β4-AR) activates adenylyl cyclase (AC) through the Gαs (Gs; s stands for stimulatory) protein [20,22,40]. However, β2- and β3-AR can also couple to Gαi (Gi; i stands for inhibitory), which abolish the effects of this enzyme [20,22,30,31,42]. AC is responsible for the intracellular generation of the second messenger cyclic adenosine 3′,5′-monophosphate (cAMP) and the activation of the cAMP-dependent protein kinase (PKA) [20,22]. PKA phosphorylates a variety of proteins regulating several cellular processes. For instance, in cardiomyocytes, PKA phosphorylates cardiac troponin I (cTnI), phospholamban (PLN), and the L-type Ca^2+^ channel (LTCC), mediating positive inotropic, lusitropic, and chronotropic effects [22]. Importantly, data in both cardiomyocytes and ECs showed that PKA downstream Gs protein activation can induce the endothelial nitric oxide (NO) synthase (eNOS), leading to the generation of the gasotransmitter NO [22,23] and subsequent activation of soluble guanylate cyclase (sGC) [22,43]. sGC is then responsible for producing cyclic guanosine 3′, 5′-monophosphate (cGMP), which stimulates the cGMP-dependent serine/threonine-protein kinase G (PKG) [22] (Figure 1). In cardiomyocytes, PKG directly phosphorylates cTnI, LTCC, PLN, and titin, then accelerates relaxation, negatively modulates contractility, and enhances the stiffness of cardiomyocytes [44]. Importantly, this complex signaling pathway confers cardioprotection, as PKG activation has been proven to reduce Ca^2+^ oscillations, which cause ventricular arrhythmias, sarcolemmal rupture, and the mitochondrial permeability transition pore (mPTP) [22,44]. In line with these data, Calvert et al. demonstrated, in vivo in mice, that activating the PKA/Akt/eNOS pathway can confer cardioprotection following ischemia/reperfusion (I/R) injury [45]. These authors showed that such molecular signaling activation was mainly related to the cardiac β3-AR subtype.

In ECs (human umbilical vein ECs (HUVECs)), the NO generated downstream from this β2-AR/PKA/eNOS system can induce potent vasorelaxant effects [46,47], and as demonstrated by our group, it can also enhance ECs function and proliferation in vitro [48]. In addition, we have also observed in a rodent model of peripheral artery disease (PAD) that the preservation and stimulation of this signaling pathway induced beneficial therapeutic effects that preserved blood flow in limbs affected by critical ischemia [47]. Finally, accordingly to Tanner et al. [49], the activation of this β2-AR/Gαs/PKA signaling pathway in CFs leads to the mitogen-activated protein kinase (MAPK) ERK 1/2 activation, resulting in CFs proliferation. However, based on their G protein-coupling characteristics, β2 and β3-AR act as brake receptors against excessive β1-AR/Gs hyperstimulation [22,50,51,52]. For example, excessive β1-AR activity may lead to cardiac arrhythmia and apoptosis, and β2- and β3-AR can counteract these effects via the Gi signaling pathway [52,53]. β1- and β2-ARs also display different effects on cardiac cell growth as stimulation of β1-AR, but not β2-AR, causes cardiomyocyte hypertrophy [54,55].

Notably, following Gi signaling pathway activation, β2- and β3-AR are able to give rise to NO with subsequent PKG activation [22]. Importantly, such an effect is mediated by the activation of eNOS or neuronal NOS (nNOS) [22] (Figure 1).

Based on this premise, a tightly regulated receptor signaling termination is essential to maintaining healthy CV system physiology and controlling the fate of each cell type. Hence, β-ARs can be rapidly inactivated by mechanisms such as those mediated by proteins called regulators of G protein signaling (RGS) that bind to the Gα subunit and stimulate their intrinsic GTPase activity (GAP activity) with subsequent hydrolysis of the active GTP-bound [42,56]. Moreover, β-ARs activation can be directly modulated by processes of phosphorylation called heterologous and homologous desensitization [20,22]. The heterologous or non-agonist specific desensitization is triggered by cAMP and diacylglycerol (DAG) that activate PKA and protein kinase C (PKC), respectively [20,22]. These PKs phosphorylate serine and threonine residues within the third intracellular loop and C-terminal (CT) tail of the β-AR. Significantly, in different cell types, such as cardiomyocytes, PKA phosphorylation of the β2-AR can induce a G protein-coupling switching from Gαs to Gαi [57].

In the homologous desensitization, which is agonist-mediated, β-AR undergoes phosphorylation by a family of serine/threonine kinases known as βARKs or GPCR kinases (GRKs), which enables β-arrestin (β-Arr1/2) recruitment with subsequent internalization of the receptor into the endosomes [20]. Importantly, GRKs present an N-terminal RGS homology (RH) domain that has been recently described as responsible for the phosphorylation independent signaling attenuation of several GPCRs, including the β3-AR [58]. Remarkably, the CT of both β1- and β2-ARs is rich in serine and threonine residues that are recognized and phosphorylated by the GRKs and present a consensus sequence for protein kinase A (PKA); the β3-AR’s CT lacks these sites [22]. Therefore, the β3-AR is more resistant than β1- and β2-AR to inactivation by homologous desensitization [22,58]. Among the GRKs identified (seven isoforms have been characterized as GRK1-GRK7), GRK2 and GRK5 represent the major isoforms expressed in the heart and vasculature, and are the primary culprits for β-AR uncoupling and dysfunction in CVD such as HF [59,60]. Indeed, due to an excessive catecholamine release and β-ARs hyperactivation, these kinases become upregulated, negatively impacting the receptors’ functionality. This mechanism is characteristic of many pathological conditions, including diabetes, endothelial dysfunction, HF, and hypertension [20,59,60,61,62,63,64]. Importantly, as discussed further in other review articles [59,60], the importance of GRK2 and GRK5 in CVD is related to their ability to induce β-AR dysregulation (canonical pathway) and to elicit toxic effects independent of receptor signaling (non-canonical pathway) [59,60,65,66,67].

## 3. Targeting β-AR Signaling in Cardiovascular Disease: GRKs Inhibition, β-Blockade and Pharmacogenomics

Based on their multiple noxious effects on β-AR signaling, the blockade of GRKs has been proposed as a potential novel therapeutic approach to fight CVDs [20,59,60]. Hence, several compounds have been developed and tested based on their structure and ability to bind and inhibit GRKs [20,60]. Among these compounds, the Takeda Pharmaceutical Company Ltd. (Osaka, Japan) developed a class of GRK2-inhibiting compounds with promising therapeutical potential [68]. Several reports from our group also analogously identified paroxetine, a serotonin reuptake inhibitor (SSRIs), authorized by the Food and Drug Administration (FDA) to treat depression and anxiety, and its derivates as highly potent and selective GRK2 inhibitors [69,70,71]. Notably, these compounds in vitro enhanced the shortening and contractility of adult cardiomyocytes in response to β-AR agonism [69,70,71]. Moreover, another FDA-approved compound with anti-inflammatory and anti-allergic immunomodulator activities has been recently identified and proposed by us as a potent GRK5-selective inhibitor through drug screening [72]. This compound, called amlexanox, in neonatal rat ventricular myocytes (NRVMs) in vitro inhibited GRK5-mediated pro-hypertrophic effects [72]. Several other molecules have been tested in recent years, including peptides derived from intra- and extra-cellular loops of the hamster β2-AR [73]. These molecules are demonstrated to antagonize GRK2 with high selectivity, interfering with receptor binding, preventing β2-AR phosphorylation and desensitization upon agonist stimulation [74]. In keeping with this line, the development of βARKct, a peptide derived from the CT of GRK2, has been undoubtedly considered a promising strategy to inhibit GRK2 with high efficiency [75,76]. Similarly to GRK2, this small peptide binds to Gβγ subunits of GPCRs, including β-ARs, reducing the capability of GRK2 to induce the phosphorylation, desensitization and downregulation of these receptors. In addition, βARKct can antagonize non-canonical and non-GPCRs related activities of GRK2. For instance, βARKct inhibits the GRK2-dependent phosphorylation of the insulin receptor substrate 1 (IRS-1), increasing glucose uptake in myocytes, and can block the mitochondrial translocation of GRK2, thus preventing myocytes cell death [20,63,73,77,78].

Among the strategies identified so far, for their ability to inhibit GRKs’ activity expression, the usage of β-blockers remains one of the most popular and effective in treating CVDs [20,79,80,81]. Indeed, Iaccarino and coworkers [82] firstly demonstrated that normal healthy mice treated chronically with isoproterenol (βAR agonist) developed myocardial hypertrophy with impaired β-AR signaling/density. These effects were associated with increased GRK2 protein levels and activity, and these authors found that atenolol (β1-AR-blocker) and carvedilol (non-selective β-AR-blocker) administration prevented GRK2 upregulation and enhanced βAR signaling and density. In line with these data, Rengo et al. [83] reported in post-ischemic HF rats that metoprolol administration significantly increased cardiac βAR density and reduced GRK2 protein levels compared to saline-treated HF controls. Thus, over the years, research in the field has fully established that β-blocker therapy preventing catecholamine hyperstimulation of β-ARs impairs GRKs expression, and leads to the restoration of β-AR density on the plasma membrane and responsiveness [84]. Despite this common mechanism, β-blockers do not all operate in the same manner. For instance, while certain β-blockers have been proved to be helpful in HF treatment, others provided no benefits or increased mortality [85]. For this reason, the drug discovery research around β-ARs developed a diverse set of pharmacological agents. This implementation was also made necessary by several factors, including the different affinity of β-blockers for each β-AR subtype, or the ability that these drugs possess to act as partial agonists for one subtype instead of another. In this context, we and others reported the ability of certain β-blockers to positively impact the expression/activity of β3-ARs [43,86,87,88,89]. For instance, Sharma and colleagues showed that β1-AR blocker metoprolol improved cardiac function in diabetic rats via β3-AR upregulation with subsequent NO generation [87]. Similarly, our group in a canine model of mitral regurgitation found that metoprolol promoted β3-AR upregulation, enhancing nNOS/NO/cGMP signaling with beneficial effects [43]. In line with these data, we recently demonstrated that metoprolol prevented β3-AR down-regulation after MI, which in turn also mediated its cardioprotective effects through the activation of the sphingosine kinase 1 (SK1) and the sphingosine-1-phosphate (S1P) pathway [69]. Further, in a model of I/R injury, Aragon and coworkers demonstrated that nebivolol, a highly selective β1-AR antagonist, activated the cardiac β3AR/eNOS/NO pathway leading to a significant reduction of infarct size [88]. These cardioprotective effects, due to nebivolol-dependent upregulation and activation of β3-AR, were similarly observed by Zhang et al. in a mouse model of myocardial infarction (MI) [89]. Interestingly, these authors demonstrated that following MI, compared to the vehicle, nebivolol treatment reduced cardiac fibrosis and apoptosis and ameliorated cardiac function. Of course, due to their presence in other cell types of the CV system, the activity of β-blockers is not limited to cardiomyocytes. For example, some of these drugs (e.g., bisoprolol, nebivolol, etc.) have also been reported to reverse endothelial dysfunction [90,91] with direct potential therapeutic effects in post-ischemic HF [92] and in hypertension [93,94,95].

Despite several shreds of evidence demonstrating the beneficial usage of β-blockers and other neurohormonal blocking strategies (e.g., GRK blockade), these cannot be considered definitive therapies [20]. One of the reasons for this statement is that not all patients respond favorably to these therapeutics, and thus, new pharmacological approaches, along with a deeper insight of the underlying molecular mechanisms contributing to the development and progression of CVD, represent the best case for future therapeutic advances [20,22,96]. Over the last few years, research around β-ARs focused on the pharmacogenomics of SNPs and their impact on HF therapeutic responses [97,98,99]. For instance, the Arg389Gly polymorphic variant of β1-AR predisposes to HF by prompting hyperactive signaling programs, leading to depressed receptor coupling and ventricular dysfunction [100]. In addition, this variant can alter responsiveness to β-blocker therapy in HF patients [101]. Conversely, the variant Ser49Gly of β1-AR was associated with myocardial protection and decreased mortality risk in patients with HF [98,99,102]. Concerning the β2-AR, Huang et al. [103] found that HF patients carrying the Arg16Gly polymorphic variant of this receptor had a worse prognosis, but responded better to β-blocker treatment than those presenting the Arg16 variant.

Other studies focused their attention on GRKs, with particular emphasis on the polymorphic variant of GRK5 Gln41Leu. Importantly this variant was previously investigated because it positively impacted the kinase activity of GRK5, protecting β-ARs against experimental catecholamine-induced cardiomyopathy, an effect called “genetic β-blockade” [104]. Interestingly, in a recent study, Ramalingam and colleagues analyzed the association of *GRK5* Gln41Leu polymorphism with response to β-blocker therapy in a cohort of Indian patients with HF, revealing that patients carrying the Leu41 variant (homozygous and heterozygous forms) presented with reduced events in hospitalization and improved cardiac output, compared to GRK5 Gln41 carriers. Moreover, these authors observed that patients with the Leu41 variant responded better to β-blocker therapy and required a lower dosage of β-blocker compared to patients with the Gln41 variant [105]. These data are in line with a study by Kang et al. [106], demonstrating, in a population of Chinese patients with systolic HF (SHF), that GRK5 Leu41polymorphism reduced the risk of SHF morbidity after β-blocker therapy, compared to the Gln41 variant.

Interestingly, previous reports found conflicting results in African Americans. Indeed, as demonstrated by Ligget et al. [104], while Leu41 carriers survived better than those presenting Gln41 variants, they exhibited a reduced response to β-blockers such as atenolol [107]. Conversely, another study among African Americans by Johnson et al. [108] demonstrated that Leu41 negatively impacted on the survival of HF patients, but these effects seemed to be stabilized upon β-blocker usage. Together, these studies suggest the importance of patients’ genetic background and ethnicity, and how this drives the proper response to β-blocker therapy. Unfortunately, despite these experimental proofs, several reports showed inconsistent results between the polymorphic variant of β-ARs and GRKs and their associated impact on β-blockers and HF outcomes [109]. Hence, other factors should be taken into account, especially considering that β-blocker therapy presents other important limitations. For example, β-blockers are not well tolerated by all patients and their usage is often associated with unwanted systemic side effects, including increased bradycardia, hypotension, weakness, dizziness, and depression [96]. Thus, choosing the appropriate β-blocker is critical for therapeutic success, and the dosage that should be titrated individually for each patient [110]. In this sense, several pharmacokinetic and pharmacodynamic studies have been performed, and differences in terms of age (young vs. older) and sex/gender (men vs. women) have been observed [111,112,113,114,115,116,117]. Importantly, these studies evaluated several factors, including the differential expression of drug-metabolizing hepatic enzymes, demonstrating that the cytochrome P450 2D6 (CYP2D6), which is involved in the metabolism of β-blockers such as metoprolol and propranolol, showed lower activity in women [111,112]. Other studies evaluated the effects of these drugs in HF with reduced ejection fraction (HFrEF), demonstrating that women, who generally have lower body weight and plasma volume, present higher plasma concentrations of β-blockers and slower clearance rates [110]. In addition, women tend to present a more significant reduction in heart rate (HR) and blood pressure (BP) than their male counterparts using similar doses [113,114]. Accordingly, women are more likely to encounter adverse and more severe drug reactions than men [113,115,116]. Therefore, it has been suggested that women with HFrEF might need lower doses of β-blockers [110]. A report from Eugene et al. [117] has also strengthened this recommendation by demonstrating gender differences in the pharmacokinetics of metoprolol in geriatric patients, concluding that the optimal dosage to be used in aged women should be almost half that of men of the same age.

## 4. Sex- and Age-Related Differences in β-Adrenergic receptors: Impact on Cardiovascular Disease

As discussed above, sex-specific differences in the pharmacokinetic or pharmacodynamic properties of therapeutic drugs such as β-blockers exist between women and men [118]. Notably, one explanation for such differences can be sought in different β-AR levels and responsiveness between men and women; these can profoundly impact the response to anti-CVD therapeutics and explain the sex differences in incidence, clinical manifestation, and outcome CVDs. For instance, in the premenopausal period, women tend to have lower BP and, therefore, a lower risk of developing hypertension than men of the same age. In this regard, in the US, the prevalence of hypertension among 20-year-old subjects was about 51.7% in men and 42.8% in women [119]. However, after menopause, women experience a much sharper incline in BP, and consequently, there is a steeper rise in the prevalence of hypertension compared to men (US adults aged 65–74 years old: 75.7% in women and 67.5% in men) [119]. Part of these effects is, at least in part, attributable to vascular β-AR responsiveness, which is higher in young women compared to age-matched men [120,121]. Concurrent with this effect, data in humans and rodents demonstrated that vessels of females constrict less and relax more than males in response to catecholamine stimulation [120,121,122,123,124,125,126]. This outcome appears to be largely dependent on β-ARs, primarily β1- and β3-AR, that limit the adverse effects of the αAR-mediated vasoconstriction [121,122]. Indeed, as demonstrated by Al-Gburi et al. [121], these receptors exhibit a higher expression in vessels of females than in males, and their selective inhibition abolished all the sex differences, equalizing both constriction and relaxation. In addition, as suggested by Riedel and coworkers [122], the vascular expression of these receptors is determined predominantly by Es, thus explaining the drop in β-AR responsiveness observed with aging in women, leading to unopposed α-adrenergic vasoconstriction and a rise in BP [127,128]. In addition, other reports have demonstrated that cardiomyocytes respond to β-AR stimulation (isoproterenol; Iso) in a sexually dimorphic manner, and this effect was associated with reduced arrhythmic activity [129,130,131]. Male cardiomyocytes show improved contractile responsiveness to Iso than female ones, predisposing the male heart to maladaptive cardiac hypertrophy development [129,130,131]. In addition to hypertrophy, male hearts are more susceptible to β-AR agonism and develop more fibrosis than females [132]. This effect is likely related to the higher β-AR expression and increased PKA activation observed in male CFs than in female counterparts [132]. However, all these effects are reverted with aging and females become more susceptible to developing CVDs. This phenomenon depends on SNS, which is hyperactivated in women in the postmenopausal period, resulting in a sustained increase in hemodynamic load that contributes to pathological functional and structural changes in blood vessels and the heart [133]. The SNS hyperactivation is associated with altered β-AR expression/signaling and then cardiac disease (e.g., HF) [19,21,134]. Indeed, postmenopausal women are more prone than men to develop cardiomyopathy called Takotsubo syndrome [135,136]. This form of cardiomyopathy is reversible and is characterized by an apical balloon appearance of the left ventricle (LV), often associated with emotional or physical stress (also called stress-induced cardiomyopathy) that leads to an increase in circulating catecholamines [137]. Previous reports demonstrated the importance of β1-AR signaling in the pathophysiology of this syndrome, as selective β1-blockade alleviated the extent of akinesia in a rat model of Takotsubo cardiomyopathy [138]. Hence, although some studies reported the application of β-blockers in Takotsubo syndrome as harmful or ineffective [139,140,141,142,143], the beneficial effects of this therapy, in terms of better long-term survival and against arrhythmia [67,144,145], have been supported. Unfortunately, the mechanisms underlying such beneficial effects still remain uncovered. However, it is possible to speculate a potential effect of β-blockers on GRK2. Indeed, Nakano et al. [146] showed that myocardial samples from patients affected by Takotsubo syndrome had higher GRK2 and β-arr2 levels than patients with dilated cardiomyopathy or healthy controls. In line with these data, Arcones et al. [147] demonstrated that a substantial increase in cardiac GRK2 levels with aging in mice could be observed only in females and not in males. Thus, as discussed above, the upregulation of GRK2 and β-arr2 activity, and downstream catecholamine stimulation (typical of aged females), represent the initial step required for β-ARs desensitization/downregulation. These data were, at least in part, corroborated by the report of Lindenfeld and coworkers [148], demonstrating that the ventricular myocardium of postmenopausal women had more pronounced β1-AR downregulation than those of younger women and young and old male counterparts (young and old).

## 5. Impact of Estrogens and Its Supplementation on β-AR Signaling: Implication in Cardiovascular Disease

In 1991, Lopez et al. [149] reported the potential role of Es in decreasing the secretion of catecholamines from the adrenal chromaffin cells, which represent an essential source of total plasma catecholamines [21]. These data were also corroborated by Park et al. [150], who showed how Es administration to the rats’ adrenal glands caused a marked inhibition of catecholamine secretion evoked by cholinergic receptor stimulation. In line with these reports, Gomes and colleagues [151] observed that, consequent to ovariectomy (OVX), female rats displayed increased plasma catecholamine levels than male rats following gonadectomy. Interestingly, these authors reported that Es supplementation was able to reverse these effects. These data imply that Es also impact β-AR expression/responsiveness. In this sense, several preclinical pieces of evidence exist, and some have recently been well-reviewed. For example, Matarese et al. [152] depicted the potential interrelation between Es and β-ARs in the mechanisms implicated in cardiac repair.

Similarly, Machuki and coworkers [153] provided an updated description of Es receptors (ERs) and β-ARs signaling, and their functional synergism in cardiac cells. However, as discussed previously by us and others, these sex hormones are crucially involved in sex differences in CVDs manifestation, clinical outcome, and response to therapy in males and females [7,8,14,152,153,154]. Notably, a significant decrease in the levels of these hormones is observed in the postmenopausal period and is considered the primary causal factor for the increased CV risk in women with aging [14]. Thus, part of the well-documented beneficial effects of Es replacement therapy (ERT) in postmenopausal women with CVD can be potentially related to the normalization of SNS and β-ARs. Indeed, Blum and coworkers [155] demonstrated that ERT resulted in a marked reduction of plasma norepinephrine (NE) in 12 postmenopausal women. In addition, as shown by Ferrer et al. [156], ERT can enhance the vasorelaxant responses induced by β-AR activation.

Unfortunately, despite the attention on novel strategies modulating β-AR signaling in CVD, few are investigating the role of ERT in humans. An explanation can be found in the reduced enthusiasm for ERT usage in postmenopausal women over the years [135]. Indeed, several clinical reports have documented various adverse effects of ERT that outweigh the benefits [157,158]. Thus, experimental and clinical studies have focused on phytoEs (PhEs), a class of plant-derived compounds with E-like activity, in addressing menopause-related disorders, including CVDs with reduced adverse effects [159,160]. Among the PhEs tested, resveratrol is a well-documented cardioprotective molecule impacting catecholamine synthesis and β-ARs. For instance, Woo et al. reported that resveratrol, perfused into an adrenal vein of normotensive rats, inhibited the acetylcholine-induced secretion of catecholamines [161]. Similar data were obtained in vitro in primary bovine adrenal medullary cells by Shinohara et al. [162] and Fernández-Morales and colleagues [163]. Notably, a consequence of this catecholamine release modulation may be responsible for the effects observed by Burstein et al. [164]. In detail, these authors demonstrated that resveratrol administration following MI in rats increased cardiac β-AR density, thus preserving cardiac contractile reserve following dobutamine administration.

## 6. Conclusions

The 1970s and 1980s were revolutionary years in CV research, as β-ARs were characterized and studied [165]. As a result, in 2012, Drs Robert Lefkowitz and Brian Kobilka, for their studies, obtained the Nobel prize for Chemistry [165]. These distinguished scientists characterized the structure of β-ARs, and the mechanisms involved in receptor desensitization [165]. After all these years, the enthusiasm around these receptors has not yet died down. Studies have revealed new signaling and molecular mechanisms through which these crucial receptors perform their actions within the cells. Importantly, as discussed in this review, three receptor subtypes (β1-, β2-, and β3-AR) have been identified so far, each with specific activities in the cells of the CV system, and their dysregulation represents a hallmark of CVD, ultimately leading to HF [20,22]. This process primarily depends on SNS hyperactivity and catecholamine bombing of β-ARs, contributing to the upregulation of GRKs that trigger β-AR downregulation and elicit non-canonical toxic activities within the cells [59,60]. Therefore, during the last three decades, more and more studies aimed at developing a pharmacological and non-pharmacological approach to counteract these effects and ameliorate the prognosis and outcome of patients with CVDs. In this context, β-blockers remain a vanguard approach since they prevent direct catecholamine toxicity and recouple β-AR to G proteins, restoring β-AR responsiveness [20]. However, the mechanisms behind β-blocker-induced protection are more complex and are not limited to simple β-AR resensitization.

Indeed, β-blockers are a heterogeneous class of drugs with different actions which remain largely uncovered. For example, certain β1- and β2-AR antagonists possess an intrinsic sympathomimetic activity (ISA) [166], a guanine nucleotide-binding regulation [167], and act as inverse agonists [168]. Thus, more studies are needed to identify new mechanisms of action behind these therapeutics. Indeed, it remains still unexplained why β-blockers have some limitations in the CVD patient population, such as the poor tolerance or non-response to this therapy. Among the vital factors to consider, the sex/gender differences are undoubtedly the most important. In this regard, the existence of sex differences in CVD risk, clinical manifestation, prognosis, and response to drug therapy between males and females is well consolidated. Moreover, as depicted in this review, these differences also involve β-AR signaling, which is differentially modulated in males and females. Most of these effects are more pronounced with aging and are related to the decline in Es levels, which also control β-AR expression and responsiveness in the CV system and modulate SNS activation. Thus, novel therapies modulating Es and then β-AR signaling are welcome. In this regard, we have discussed the potential of ERT and PhEs that, behind their effects, appear to impact SNS activity and β-AR signaling in postmenopausal women (Figure 2). Unfortunately, despite the tremendous emerging interest in this topic, females remain poorly represented in preclinical and clinical studies. For these reasons, several scientists started dissecting the effects of anti-CVDs therapies in men and women and evaluating the different responses and outcomes. In this sense, Bugiardini et al. [169] recently analyzed the association between the use of β-blocker therapy in women with hypertension and the increased risk of developing HF. Therefore, future studies dissecting the different responses to specific treatments in men and women will provide further actionable data that will help to optimize diagnostic and therapeutic strategies for treating CVDs in a sex-dependent manner, which is the concept of “sex/gender CV medicine”.

## Figures and Tables

**Figure 1 jcm-11-04280-f001:**
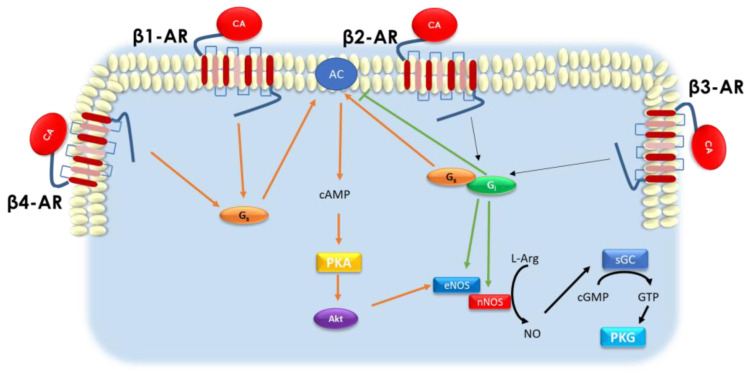
**Schematic representation of β****-adrenergic receptor (AR)-mediated induction of protein kinase A (PKA) and PKG downstream Gs and Gi protein activation**. β1-, β2-, β3-, and the putative β4-AR are coupled to the stimulatory G proteins (Gs), while only β2- and β3-AR are also coupled to the inhibitory G proteins (Gi). Upon ligand binding (catecholamine, CA), Gs proteins activate the adenylate cyclase (AC) on the plasma membrane leading to the generation of Cyclic Adenosine Monophosphate (cAMP), with subsequent activation of PKA, that in turn phosphorylates several key factors, including the PKB (Akt), with the subsequent activation of the endothelial nitric oxide synthase (eNOS). Of note, eNOS activation increases the generation of NO that stimulates the soluble guanylate cyclase (sGC) to produce cGMP and PKG activation. Notably, following the Gi signaling pathway activation, β2- and β3-AR can give rise to NO via both eNOS and neuronal NOS (nNOS), thus leading to PKG induction. Gs—pathway in orange; Gi—pathway in green. Black arrows: Gs and Gi pathway.

**Figure 2 jcm-11-04280-f002:**
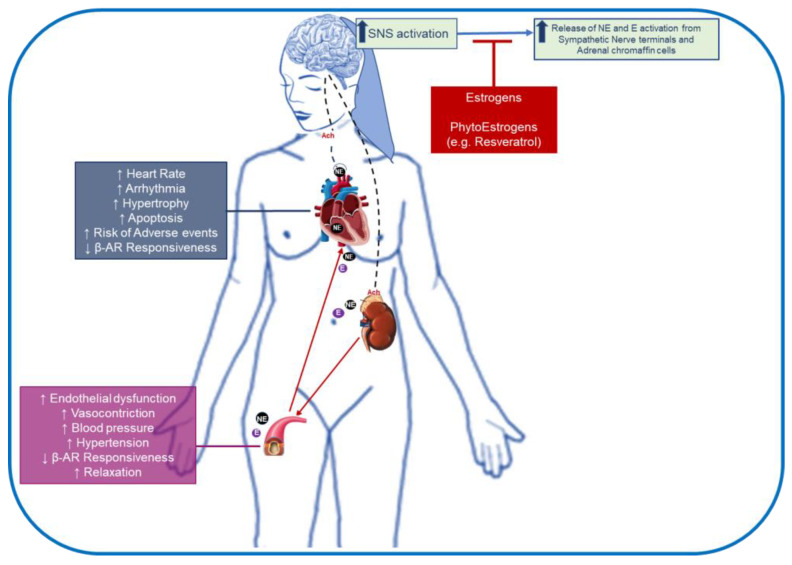
**Potential effects of estrogen or phytoestrogen therapy on sympathetic nervous system (SNS) hyperactivity in postmenopausal women**. Increased SNS activity in postmenopausal women induces catecholamine secretion from the chromaffin cells of the adrenal medulla (~80% epinephrine (E) and 20% norepinephrine [NE]), or the postganglionic sympathetic fibers (~80% NE and 20% E), resulting in increased circulating catecholamine levels, accelerating the risk of adverse vascular and cardiac effects. Estrogen or phytoestrogen therapies may inhibit SNS overactivation, preserving β-adrenergic receptors (β-Ars) responsiveness and density, and therefore ameliorating most of the cardiovascular disease (CVD) risk factors.

## Data Availability

Not applicable.

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
