# Peer review of "Sex/Gender- and Age-Related Differences in β-Adrenergic Receptor Signaling in Cardiovascular Diseases"

_jcm, 2022, doi:10.3390/jcm11154280_

Round 1

Reviewer 1 Report

The manuscript by Liccardo et al is entitled Sex/Gender- and Age-related differences in β-adrenergic Receptor Signaling in Cardiovascular Diseases  

It is well written in general but it lacks from clarity in some parts. My main concern is that the title doesn’t represent what is in the manuscript. The manuscript is more clearly based on the description of the mechanisms underlying β-adrenergic receptor activation rather than on the differences found in drug therapies between women and men. Moreover, only differences in the effects of beta-blockers between gender are evoked (apart from one mention of β-AR levels in male CFs l293 and GRK2 levels in aged females l315). The manuscript is divided into 5 parts including the introduction. The differences in gender and age are evoked only from the middle of the 4th part, and the last part mentions estrogen supplementation, so the description of what is observed in women compared to men represents only a small proportion of the manuscript compared to the mechanisms. That should be much more developed as it is supposed to be the basis of the review. In conclusion, the mechanisms should be reduced and the part on sex differences (and especially in association with aging) should be extended.

In addition, many reviews (Herz 2002, Madamanchni 2007, Oliver et al 2019, …) already focused on the mechanisms underlying beta-adrenergic receptor effects, and even though they are extremely detailed here, I don’t see new pieces of information. Again I don’t see the point in giving a detailed description of pathways if that is not linked with a difference between women and men.

In detail,

Part 2: the second paragraph (l99-l135) is very confused and should be rewritten. It has clearly been re-edited a few times. As an example, it is unclear whether β2-AR/PKA works through Gi or Gs stimulation, and if β3-AR/PKG involves only Gi? I am not sure to see the point of the detailed description of these mechanisms in terms of differences in strategies or effects of drugs between genders…

Also, there is a repetition of the mention of NO production through AC stimulation (l101 and 113). Same as the mention of PKA as it was the first time (l131). GRKs is described l143 but was already mentioned l136…

Part 3: the title is not suitable as it describes GRKs more than β-ARs targeting. I don’t understand the sentence l183. How a beta-blocker interferes with GRKs?? Also, I don’t understand l194-195, please explain.

Part 4: the title is not appropriate. If I am not wrong, it is more on the altered effects of therapies due to genetic variations and gender, rather than on the differences in β-AR? In addition, why do the authors give so many details here on genetic variations of β-Ars when there has nothing to do with gender??? L234-238: what is the link between “conflicting results about beta-blockers” and the following mention of only one study showing a positive effect of a GRK5 variant? This paragraph should be rewritten to better explain how variations of beta-receptors and/or GRKs may modify the effect of beta-blockers. By the way, please explain why a genetic modification in GRKs may influence the effect of beta-blockers?? What is the point of l241-245? The mention of differences in the effect of the beta-blockers considering age and gender appears to be very late in the manuscript…

Part 5: the title should indicate Estrogen supplementation

Minor:

L67: should be similar to other titles (bold)

L 85. A dot is missing after the coma

L98: replace “opposing” with “different” effects (one causes hypertrophy but the other has no effect- doesn’t induce hypotrophy)

L179: remove “however”

L301: Takotsubo was mentioned before, without a description…

L334: this review is exactly what was supposed to be in this manuscript!

Author Response

Reviewer 2.

  1. The manuscript by Liccardo et al. is entitled Sex/Gender- and Age-related differences in β-adrenergic Receptor Signaling in Cardiovascular Diseases. It is well written in general, but it lacks from clarity in some parts. My main concern is that the title doesn’t represent what is in the manuscript. The manuscript is more clearly based on the description of the mechanisms underlying β-adrenergic receptor activation rather than on the differences found in drug therapies between women and men. Moreover, only differences in the effects of beta-blockers between gender are evoked (apart from one mention of β-AR levels in male CFs l293 and GRK2 levels in aged females l315). The manuscript is divided into 5 parts including the introduction. The differences in gender and age are evoked only from the middle of the 4thpart, and the last part mentions estrogen supplementation, so the description of what is observed in women compared to men represents only a small proportion of the manuscript compared to the mechanisms. That should be much more developed as it is supposed to be the basis of the review. In conclusion, the mechanisms should be reduced and the part on sex differences (and especially in association with aging) should be extended. In addition, many reviews (Herz 2002, Madamanchni 2007, Oliver et al 2019, …) already focused on the mechanisms underlying beta-adrenergic receptor effects, and even though they are extremely detailed here, I don’t see new pieces of information. Again, I don’t see the point in giving a detailed description of pathways if that is not linked with a difference between women and men.

Reply: We thank the Reviewer for His/Her constructive feedback and comments on our manuscript. As discussed by the Reviewer, several studies (both reviews and original articles) depicted the role of β-ARs in the CV system and disease. Hence, we are very thankful that with this round of Revision, we can discuss and reference more exciting articles like those provided here that will increase the overall interest and impact of our Review article (please see page lines). Regarding the Reviewer's statement, "I don't see new pieces of information," we respectfully argue that the Review is the first to illustrate and put together almost all the knowledge about differences between women and men in β-AR signaling in the CV system. In addition, in our Review, we did not focus only on the differences in the heart (like two recent fascinating review articles by Matarese et al. 2021 and Machuki et al. 2018). However, we also explored the effects in other cell types like cardiac fibroblasts, endothelium, and vascular smooth muscle cells, and to our knowledge, no one has reviewed this literature before. It is worth noting that this research was complex because few original studies, especially in recent years, explored sex differences related to βAR in the CV system, and we think to have referenced almost all of them. We have revised the manuscript to address all the points raised. We trust that the responses and corresponding changes made to the revised manuscript, as detailed below, have satisfactorily addressed all the issues and that our revised manuscript will now be deemed suitable for publication in the Journal of Clinical Medicine.

Major

  1. Part 2: the second paragraph (l99-l135) is very confused and should be rewritten.

Reply: The paragraph has been completely revised. Thanks.

  1. It has clearly been re-edited a few times. As an example, it is unclear whether β2-AR/PKA works through Gi or Gs stimulation, and if β3-AR/PKG involves only Gi?

Reply: We apologize for the confounding information. As per our previous reply, we have re-edited the whole paragraph according to the Reviewer’s suggestion, and regarding this point, we specify that “L91 page The specific function and signaling activated by each β-AR in the CV system are primarily defined by the Gα-subunit to whom there are coupled. In this regard, stimulation of all β-ARs (β1-, β2-, β3- and the putative β4-AR) activates adenylyl cyclase (AC) through the Gαs (Gs; s stands for stimulatory) protein [20, 22, 40]. However, β2- and β3-AR can also couple to Gαi (Gi; i stands for inhibitory), which abolish the effects of this enzyme [20, 22, 30, 31, 42]. ……L105 Importantly, data in both cardiomyocytes and ECs showed that PKA downstream Gs protein activation can induce the endothelial nitric oxide (NO) synthase (eNOS), leading to the generation of the gasotransmitter NO [22, 23] and subsequent activation of soluble guanylate cyclase (sGC) [22, 43]. sGC is then responsible for producing cyclic guanosine 3', 5'-monophosphate (cGMP), which stimulates the cGMP-dependent ser-ine/threonine-protein kinase G (PKG) [22] (Figure 1). …….. L157 Notably, following Gi signaling pathway activation, β2- and β3-AR are able to give rise to NO with subsequent PKG activation [22]. Importantly, such effect is mediated by the activation of eNOS or neuronal NOS (nNOS) [22](Figure 1).”. Therefore, as described in this paragraph, β3-AR/PKG involves both Gs and Gi pathways. This mechanism has been illustrated in the new Figure 1. Thanks

  1. Also, there is a repetition of the mention of NO production through AC stimulation (l101 and 113).

Reply: We have revised this point accordingly.

  1. Same as the mention of PKA as it was the first time (l131).

Reply: We have revised this point accordingly.

  1. GRKs is described l143 but was already mentioned l136…

Reply: We have revised this point accordingly.

  1. Part 3: the title is not suitable as it describes GRKs more than β-ARs targeting.

Reply: This is another good point. We have changed the title to “Targeting β-AR signaling in cardiovascular disease: GRKs inhibition, β-blockade and pharmacogenomics”. Thanks

  1. I don’t understand the sentence l183. How a beta-blocker interferes with GRKs??

Reply: We thank the Reviewer for raising this concern. We have again revised the whole paragraph. Regarding this point, we have clarified how beta-blockers counteract GRKs. Please see page 6 L255-273.

  1. Also, I don’t understand l194-195, please explain.

Reply: We thank the Reviewer for raising this concern. We have again revised the whole paragraph 3. Please see pages 5 and 6 L242-278

  1. Part 4: the title is not appropriate. If I am not wrong, it is more on the altered effects of therapies due to genetic variations and gender, rather than on the differences in β-AR?

Reply: We apologize for the confusion. We moved the first part of the old part 4 (from L224 to L245) to part 3. See page 6 and 7 L299-337.   

  1. In addition, why do the authors give so many details here on genetic variations of β-Ars when there has nothing to do with gender???

Reply: We thank the Reviewer for raising this concern. We agree that, in general, the role of genetic variation may not appear relevant to the general concept of gender and BAR signaling. However, although briefly, We had to include this critical argument in part 3 of our manuscript to discuss all the potential reasons why patients with CVDs respond favorably to β-blockers. We hope that the Reviewer will agree with our point of view.

  1. L234-238: what is the link between “conflicting results about beta-blockers” and the following mention of only one study showing a positive effect of a GRK5 variant? This paragraph should be rewritten to better explain how variations of beta-receptors and/or GRKs may modify the effect of beta-blockers. By the way, please explain why a genetic modification in GRKs may influence the effect of beta-blockers??

Reply: We thank the Reviewer for raising such a significant concern. Accordingly, we have moved the discussion about how genetic impact beta-blockers to part 3 of our manuscript. Please see page 7 L316-337

  1. What is the point of l241-245? The mention of differences in the effect of the beta-blockers considering age and gender appears to be very late in the manuscript…

Reply: We agree with the Reviewer in full. For this reason, we moved the point of l241-245 to part 3. Please see page7 L339-336

  1. Part 5: the title should indicate Estrogen supplementation

Reply: We have changed the title to “Impact of Estrogens and its supplementation on β-AR signaling: implication in cardiovascular disease”. Thanks

Minor:

  1. L67: should be similar to other titles (bold)

Reply: Thanks, we have corrected accordingly.

  1. L 85. A dot is missing after the coma

Reply: We have revised accordingly. Page line. Thanks

  1. L98: replace “opposing” with “different” effects (one causes hypertrophy but the other has no effect- doesn’t induce hypotrophy)

Reply: We have revised accordingly. Page line. Thanks

  1. L179: remove “however”

Reply: We have revised accordingly. Page line. Thanks

  1. L301: Takotsubo was mentioned before, without a description…

Reply: We have removed the previous mention about takotsubo syndrome and discussed it in the paragraph 4. Page and line. Thanks

  1. L334: this review is exactly what was supposed to be in this manuscript!

Reply: We agree with the Reviewer that both the Review articles from and Matarese et al. and Machuki et al. well-illustrated the sex differences in BAR signaling. However, as discussed above in reply to point 1, these authors depicted the interrelation between Estrogens or ERs and β-ARs in the cardiac setting. With the present Review article, we have broadened the discussion not only focusing on estrogens (a thematic of part 5 of our manuscript) and their supplementation on BAR signaling, but in part 4, we discussed all the sex differences in the CV system between men and women in terms of mere BAR signaling. Of course, we recognize that our original version of the manuscript resulted in some points confusing and not directly focused on the central thematic, but thanks to this revision, we think that now our manuscript has reached the priority to be approved for publication.

Reviewer 2 Report

The authors provide a very interesting and comprehensive review of the selected issue. The review includes all relevant works on this topic in recent years, as well as the results of the authors' own research.The manuscript should be recommended for publication in its current version.

Author Response

Reviewer 4.

The authors provide a very interesting and comprehensive review of the selected issue. The review includes all relevant works on this topic in recent years, as well as the results of the authors' own research.The manuscript should be recommended for publication in its current version.

Reply: We thank the Reviewer for her/his appreciation of our work.

Reviewer 3 Report

The article presents the relationship of adrenergic receptors with sex and age. The authors describe the CV diseases in which beta-adrenergic receptors participate and their relationship with estrogens, sex differences in expression, signaling between subtypes, their relationship with the nitric oxide pathway, and describe desensitization. It also describes the use of GRK inhibitors as well as SNPs in the response to beta adrenergics.

The authors do not consider beta-4 receptors and how they are associated with cardiovascular responses.

Page 2:

Line 67: it is necessary to correct the title is not in the format

Line 68-73: does not describe the difference in expression to which mechanism they are associated at the level of genes

Line 83: Specify which subtype of beta receptors are coupled to Gs

It is necessary to check the abbreviations for example HF because they are not all defined

Page 3:

No comment

Page 4:

No comment

Page 5:

No comment

Page 6:

No comment

Page 7:

No comment

Page 8:

No comment

Page 9:

References some have the DOI and others don't have it

Author Response

Reviewer 5.

The article presents the relationship of adrenergic receptors with sex and age. The authors describe the CV diseases in which beta-adrenergic receptors participate and their relationship with estrogens, sex differences in expression, signaling between subtypes, their relationship with the nitric oxide pathway, and describe desensitization. It also describes the use of GRK inhibitors as well as SNPs in the response to beta adrenergics.

Reply: We thank the Reviewer for His/Her constructive feedback and comments on our manuscript. We have revised the manuscript to address all the points raised. We trust that the responses and corresponding changes made to the revised manuscript, as detailed below, have satisfactorily addressed all the issues and that our revised manuscript will now be deemed suitable for publication in the Journal of Clinical Medicine.

The authors do not consider beta-4 receptors and how they are associated with cardiovascular responses.

Reply: We thank the Reviewer for highlighting this critical issue to our attention. In the revised version of our manuscript, we include the putative existence of the beta-4 receptors and their cardiac role. Please see page 2 L75-83 and L93-94. 

Page 2:

  1. Line 67: it is necessary to correct the title is not in the format

Reply: Thanks, we have corrected it accordingly.

  1. Line 68-73: does not describe the difference in expression to which mechanism they are associated at the level of genes

Reply: We have revised the paragraph to insert the info related to the B4-AR and added this information in the text (please see. Page 2 L69-70). Thanks.

  1. Line 83: Specify which subtype of beta receptors are coupled to Gs

Reply: We have added this information in the text (please see. Page 2 L93-94). Thanks.

  1. It is necessary to check the abbreviations for example HF because they are not all defined

Reply: We have added this information in the text (please see Page2  L58.). Thanks.

Page 9:

  1. References some have the DOI and others don't have it

Reply: The reference list has been revised. Thanks.

Round 2

Reviewer 1 Report

A substantial reorganization of the manuscript has been made, which adds clarity.